# The Chemokine CXCL14 as a Potential Immunotherapeutic Agent for Cancer Therapy

**DOI:** 10.3390/v16020302

**Published:** 2024-02-16

**Authors:** Nicholas S. Giacobbi, Shreya Mullapudi, Harrison Nabors, Dohun Pyeon

**Affiliations:** Department of Microbiology and Molecular Genetics, Michigan State University, East Lansing, MI 48824, USA; giacobbi@msu.edu (N.S.G.); mullapu1@msu.edu (S.M.); naborsha@msu.edu (H.N.)

**Keywords:** CXCL14, immunotherapy, chemokine

## Abstract

There is great enthusiasm toward the development of novel immunotherapies for the treatment of cancer, and given their roles in immune system regulation, chemokines stand out as promising candidates for use in new cancer therapies. Many previous studies have shown how chemokine signaling pathways could be targeted to halt cancer progression. We and others have revealed that the chemokine CXCL14 promotes antitumor immune responses, suggesting that CXCL14 may be effective for cancer immunotherapy. However, it is still unknown what mechanism governs CXCL14-mediated antitumor activity, how to deliver CXCL14, what dose to apply, and what combinations with existing therapy may boost antitumor immune responses in cancer patients. Here, we provide updates on the role of CXCL14 in cancer progression and discuss the potential development and application of CXCL14 as an immunotherapeutic agent.

## 1. Tumor-Suppressive and Tumor-Promoting Roles of Chemokines in Cancer

Chemokine signaling is part of the unique language in which the immune system communicates with itself and other cell types. Chemokines were initially identified as chemical attractants for immune cells but were later discovered to modulate a broad range of homeostatic functions and responses to pathogens and diseases [1,2,3]. Chemokines also play important roles in cancer progression and the modeling of the tumor microenvironment (TME) [4]. A recent comprehensive profiling of the chemokine signaling governing the TME was shown in the study by Hoch et al. through their examination of patient-derived melanoma [5]. Hoch et al. have shown that so-called immunologically “cold” tumors lack chemokine expression and T-cell infiltration. In contrast, immune “hot” tumors show high T-cell infiltration within areas exhibiting relatively high levels of the chemokines CXCL9, CXCL10, CXCL13, and CCL4 [5]. The role of CXCL9 and CXCL10 in T-cell infiltration has been well established, representing their importance for tumor clearance by T-cell responses [6,7,8,9,10,11]. The B-cell homing chemokine, CXCL13 [12], was newly implicated in ovarian cancer by mediating the formation of tertiary lymphoid structures and infiltrating T and B cells into tumors [13]. Galeano Niño et al. have also shown that CD8+ T cells within the TME induce the recruitment of distant CD8+ T cells via the expression of CCL3/4, establish a positive feedback loop, and eventually “swarm” the tumor with CD8+ T cells [14]. While these studies suggest the importance of chemokines in activating and perpetuating antitumor immune responses, chemokines can also promote the opposite effect, establishing a protumor and immunosuppressed TME. For example, Li et al. have shown that activating the CCL2-CCR2 axis results in the recruitment of myeloid-derived suppressor cells and tumor-associated macrophages (TAMs) into the TME, leading to immunosuppression and tumor promotion [15]. Similarly, Xu et al. have shown that TAMs upregulate CCL5 expression in renal cell carcinoma, creating an immunosuppressive TME associated with poor patient prognosis [16]. Together, these studies indicate that chemokines have the capacity to both enhance and negate antitumor immune responses. Identifying the most applicable conditions and relevant chemokine(s) in the right context (e.g., cancer type) may offer the potential to utilize chemokines as novel cancer immunotherapeutics.

## 2. Utilizing Chemokines to Augment Immunotherapy 

The goal of cancer immunotherapy is to activate antitumor immune responses. To date, the use of chemokines has shown limited efficacy as monotherapies [4,17]. However, given their critical functions in immune cell regulation, chemokines could be used for effective combination immunotherapies [18]. Indeed, some recent studies have shown that the response to immune checkpoint inhibitors (ICIs) targeting PD-1 and PD-L1 relies on T-cell recruitment by CXCL9 and CXCL10 expression and CXCR3 signaling [19,20,21]. Inversely, the Combination of BL-8040 and Pembrolizumab in Patients with Metastatic Pancreatic Cancer (COMBAT) clinical trial, which examined a CXCR4 antagonist (BL-8040) in combination with a PD-1 inhibitor (pembrolizumab) and chemotherapy, exhibited promising results in treating pancreatic ductal adenocarcinoma [22,23], demonstrating that the negation of specific chemokine signaling may also serve as a viable strategy for effective immunotherapy.

Using chemokines has shown an important impact on chimeric antigen receptor (CAR) T-cell therapy. While CAR T-cell therapies have been effective in treating some leukemias, lymphomas, and myelomas, they are less effective in treating solid tumors due to limited CAR T-cell infiltration into the TME [24,25,26]. Thus, attracting T cells by chemokines has been suggested to overcome this limitation of CAR T-cell therapy. For example, Wang et al. utilized an adenoviral vector to express CXCL11 [26], which recruits T cells into peripheral tissues [27]. As T cells express high levels of CXCR3, a CXCL11 receptor, CXCL11 could be used as a strong chemoattractant to recruit CAR T cells into the TME. Ultimately, Wang et al. have shown that CAR T-cell therapy alone is ineffective in inhibiting tumor growth and requires the addition of CXCL11 to achieve a significant antitumor response [26]. 

It is important to reiterate that although chemokines (or their blockades) can be combined with ICIs or CAR T-cell therapies, an unfettered application of chemokines across situations may be ineffective or even harmful. Thus, prior evidence consistent with an antitumor response and safety in each context should be necessary for selecting chemokines and preparing regimens for treatment.

## 3. CXCL14 and Cancer 

CXCL14, a homeostatic chemokine in squamous epithelia, is known for its association with cancer as being abundantly expressed in normal tissue but significantly downregulated in some tumors [28,29]. High levels of CXCL14 expression are correlated with overall patient survival in colorectal, breast, endometrial, intraepithelial, and head and neck cancers [30] and suppress tumor progression [31,32,33,34,35]. In contrast, other studies have shown protumor effects of CXCL14 in nasopharyngeal carcinoma, prostate cancer, glioblastoma, non-small-cell lung cancer, and microsatellite-stable colorectal tumors [25,36,37,38,39]. 

We have previously shown that CXCL14 expression is epigenetically downregulated in human papillomavirus-positive (HPV+) head and neck squamous cell carcinoma (HNSCC) and cervical cancer (CxCa) [40,41]. By rescuing the expression of CXCL14 in HPV+ HNSCC cells, we observed increased MHC-I expression and CD8+ T cell infiltration into the TME [41,42], resulting in tumor suppression in vivo. Our findings suggest that CXCL14 is critical for the antitumor control of HPV+ cancers. Similarly, Kumar et al. have shown that elevated CXCL14 resulted in increased CD8+ T-cell infiltration into tumors with improved survival using an in vivo malignant glioma model [32]. Dolinska et al. have shown that CXCL14 expression is absent in bone marrow niche cells of chronic myeloid leukemia patients. However, with CXCL14 restoration, leukemia-initiating stem cells were suppressed, and their sensitivity to imatinib treatment was enhanced [42,43]. Parikh et al. have revealed that CXCL14 expression inhibits tumor growth and increases tumor-infiltrating lymphocytes in HPV-negative squamous-cell carcinoma of the oral cavity [44]. Interestingly, single cell-RNA sequencing has revealed that CXCL14 downregulation is most prominent in malignant cells within tumor-draining lymph nodes as well as the primary tumor cells, suggesting that CXCL14 may play an important role in limiting nodal metastasis [44]. The effect of CXCL14 on metastasis has previously been observed in other cancer types [30,42,45,46]. Conversely, metastasis enhanced by CXCL14 has been shown in pancreatic and breast cancers [30,47,48]. Overall, these findings suggest CXCL14 has diverse and likely context-specific functions in antitumor immunity and metastasis [30,35,49,50].

To better understand the purview of CXCL14 functions and potential interactions with biological pathways in cancer progression, we performed gene-to-function analyses using Ingenuity Pathway Analysis (IPA) [51]. We then further categorized the IPA results of increased functions (blue segments) and decreased functions (red segments) into four broad groups (Figure 1). As expected, the most documented functions that are enhanced in the wake of increased CXCL14 expression are related to cellular immunity, including lymphocyte chemotaxis and tumor suppression. Interestingly, the documented inhibitory functions can be categorized together under cancer progression. For example, the findings of Tessema et al. [34] and Izukuri et al. [52] demonstrate the functions of CXCL14 in suppressing cancer progression. Tessema et al. have shown that the *CXCL14* promoter is hypermethylated in lung adenocarcinoma cells and that restored CXCL14 expression induces the necrosis of xenografted lung adenocarcinoma [34]. Izukuri et al. have revealed that CXCL14 transgenic mice with high CXCL14 levels in the blood limit intratumoral neovascularization and suppress the progression of lung adenocarcinoma and melanoma [52]. Additionally, CXCL14 may also be involved in a tumor-progressing metabolic mechanism, such as the uptake of glucose, which was observed to be inhibited by CXCL14 (Figure 1) [53]. Although it is unknown whether the CXCL14-mediated inhibition of 2-deoxyglucose uptake is linked to cancer progression, it may be plausible that CXCL14 suppresses cancer progression by limiting glucose uptake in cancer cells when considering the reliance of cancer cells on glucose as part of the Warburg effect [54]. Overall, our findings from the pathway analysis provide insights into the future directions of investigating the role of CXCL14 in antitumor immunity and cancer progression. 

## 4. Developing CXCL14 as an Immunotherapeutic Agent 

Based on the antitumor activity of CXCL14 through immune activation in several cancers [32,40,42,43,45,65,66], we propose that CXCL14 could be used in cancer immunotherapy, particularly in treating HPV+ cancers [40,42]. This possibility is further supported by the association of high CXCL14 expression with better patient survival in multiple cancers, including HNSCC and CxCa [41]. Furthermore, new evidence from Pan et al. has shown that CXCL14 expression is associated with an enhanced response to immunotherapy in renal cell carcinoma [67]. Additionally, CXCL14 has several beneficial features as a therapeutic agent. CXCL14 is a small soluble protein with a size of ~10 kDa and can be delivered by several different vehicles, such as viral vectors, liposomes, and nanoparticles. Because CXCL14 is constitutively expressed by many cell types throughout the body for maintaining homeostasis [30], it is less likely to trigger any adversary effects commonly caused by proinflammatory chemokines [68]. This could be a key advantage of using CXCL14 compared to other cytokines (e.g., IL-2), many of which have been shown to cause high toxicity or trigger immune suppression via regulatory T cells [69]. Lastly, CXCL14 has dual functions in enhancing antitumor responses by recruiting natural killer (NK) and T cells into the TME and upregulating MHC-I expression on tumor cells to enhance antigen presentation [40,42,67]. Thus, CXCL14 has great potential as an immunotherapeutic agent in novel combination immunotherapy with an ICI, particularly for ICI nonresponders. To achieve this goal, we need to better understand the mechanism of CXCL14-mediated tumor suppression and further develop effective delivery methods.

## 5. Understanding the Mechanism of CXCL14-Mediated Tumor Suppression 

We have previously shown that CXCL14 upregulates MHC-I expression in HPV+ HNSCC cells and increases the infiltration of NK, CD4+, and CD8+ T cells into the TME [41,42]. However, the native CXCL14 receptor(s) and their signaling pathways required for MHC-I upregulation and immune cell chemotaxis remain to be elucidated. Tanegashima et al. have shown that CXCL14 binds to CpG oligodeoxynucleotides (ODNs) to induce toll-like receptor 9 (TLR9) signaling in dendritic cells [70]. They further identified that the N-terminal loop structure of CXCL14, as distinct from other CXC chemokines, is required for DNA recognition and internalization [71]. TLR9 activation also requires the N-terminal domain at amino acids 1–12 and the 40S loop at amino acids 41–47 [71]. 

The evidence related to TLR9:CXCL14 signaling may also explain the orphan receptor status of CXCL14 with its activity outside the paradigm of typical CXC-CXCR receptor interactions. On the other hand, Witte et al. have shown that CXCL14 binds to CXCR4 to mediate platelet and monocyte chemotaxis [72]. However, the possibility of CXCR4 as the receptor for CXCL14 is controversial without any definitive conclusion [73,74,75,76]. Despite these findings, it is still unclear whether TLR9 and/or CXCR4 are involved in MHC-I upregulation and/or NK and T cell recruitment by CXCL14. Although CXCR4 is highly expressed on T cells and important for T cell chemotaxis [77], there is no evidence that the CXCL14-CXCR4 axis plays any role in T cell recruitment to the TME. Kouzeli et al. have shown that CXCL14 synergistically enhances interactions of CXCL13, CCL19, and CCL21 with their receptors to increase immune cell chemotaxis in vitro [78]. Thus, it is possible that CXCL14 plays a broad and non-linear role in MHC-I upregulation and lymphocyte chemotaxis beyond a direct ligand-receptor interaction.

## 6. Developing Effective Delivery Tools for CXCL14

A pharmacologic approach to restore CXCL14 may be possible using the DNA methyltransferase inhibitor (DNMTi) decitabine (5-aza-2’-deoxycytidine). We have previously shown that decitabine treatment reverses *CXCL14* promoter hypermethylation and upregulates *CXCL14* expression in HPV+ cancer cells [42]. Other groups have also shown the effectiveness of decitabine in upregulating CXCL14 to treat cancer [34,66,79]. However, the expression of many genes is affected by DNA methylation. Given the global impact on DNA methylation by decitabine, treatment may cause undesired effects and/or drug-related toxicity in patients beyond restoring CXCL14 expression. Thus, developing methods for the ectopic delivery of CXCL14 may be necessary to use CXCL14 as an immunotherapeutic agent. 

Potential methods for CXCL14 delivery include the administration of recombinant CXCL14 protein directly into the TME and *CXCL14* gene using nanoparticles, liposomes, or viral vectors. Because each method has clear advantages and disadvantages, as previously documented [80,81,82,83], effective CXCL14 delivery may depend on these technical limitations and patients’ circumstances (e.g., immunocompromised condition). For instance, the direct application of recombinant CXCL14 protein may be challenged by the lack of posttranslational modifications required for its proper functions [28,84]. Nanoparticles and liposomes have been shown to be effective in previous cancer therapies but are still limited by their route of administration and biodistribution [85]. On the other hand, viral vectors, such as adenovirus and vaccinia virus, have been shown to effectively deliver chemokine genes and boost antitumor immune responses as an additional benefit, given their documented role in innate immune activation [86,87]. Conversely, pre-existing immunity can quickly eliminate these viruses and block successful CXCL14 delivery [88]. Thus, further tests are warranted to determine the most effective methods for CXCL14 delivery.

## 7. Optimizing CXCL14 Protein Stability

To develop an effective CXCL14 immunotherapy, the protein stability of CXCL14 (and chemokines generally) needs to be improved due to the short half-life of chemokines [89]. Based on the previous study reporting the degrons, the amino acid motifs that facilitate protein degradation, we hypothesized that the deletion of two consecutive glutamates (CXCL14-dEE) at the carboxy-terminus stabilizes CXCL14 protein [30,90]. To test the hypothesis, we first analyzed the structure of CXCL14-dEE compared to wildtype CXCL14 (CXCL14-WT), using AlphaFold and SWISS-MODEL. The in silico analysis of the CXCL14-dEE structure did not show any significant alterations in protein folding or tertiary conformation compared to CXCL14-WT, except only minor changes from the truncation of two glutamates (Figure 2). 

To examine the functional changes by deleting the two glutamates, we performed cycloheximide (CHX) chase assays with MG-132-treated 293T cells expressing CXCL14-WT or CXCL14-dEE. Replacing MG-132 with CHX permits the evaluation of protein stability over time by inhibiting de novo protein synthesis (Figure 3A). MG-132 treatment does not increase levels of CXCL14-dEE to match that of CXCL14-WT, and subsequent CHX treatment showed a greatly reduced half-life of CXCL14-dEE. This result suggests that CXCL14-dEE is significantly less stable than CXCL14-WT. Additionally, Peterson et al. have shown that mutations in the “destruction-box” at arginine-43 and tyrosine-44 in CXCL14 stabilize CXCL14 protein by eliminating the E3 ligase recognition site (CXCL14-RY43/44AA) [98]. As expected, our in silico prediction of CXCL14-RY43/44AA protein structure showed no major changes compared to CXCL14-WT (Figure 3A). CHX treatment significantly enhanced the protein stability of CXCL14-RY43/44AA compared to CXCL14-WT, which is consistent with the previous result [98]. Interestingly, however, when cells were treated with CHX or MG-132 alone, CXCL14-RY43/44AA showed modest but consistently higher protein levels relative to CXCL14-WT. Additionally, chase experiments with brefeldin A treatment showed that CXCL14-RY43/44AA accumulated significantly faster than CXCL14-WT (Figure 3B). These results suggest that the RY43/44AA mutation could contribute to increasing intracellular CXCL14 levels by limiting its secretion, but this possibility should be confirmed by further investigation. 

## 8. CXCL14 Therapeutic Dose and Combination Therapy

Determining the CXCL14 levels required for inducing effective antitumor immunity while being safe for a patient to receive is crucial for the success of using a CXCL14 immunotherapeutic agent. We have previously shown that restoring the physiological levels of CXCL14 in HPV+ HNSCC cells significantly suppresses tumor growth in immunocompetent syngeneic mice [42]. Our results firmly support the notion of CXCL14-stimulated antitumor immunity for treating HPV+ HNSCC. However, the optimal therapeutic dose of CXCL14 is still unknown. Furthermore, given that tumorigenic mechanisms can be unique to each patient and coupled with the elaborate complexities of individual TMEs [100,101], the effective level of CXCL14 required for individual treatment may vary significantly among different patients. Even if therapeutic levels of CXCL14 could be clinically attained, the local application of CXCL14 to (or from) a single tumor site may not instigate an adequate immune response to promote cancer clearance in patients with advanced metastatic disease. This point is especially relevant in HPV+ HNSCC, where ~50% of cases have an unidentified primary tumor, and cancer is only discovered following the detection of metastatic tumors [102]. Although some studies have indicated that high systemic levels of CXCL14 are safe in humans or mice [35,52], applying CXCL14 systemically may be impractical and still have unforeseen negative consequences. Thus, the local administration of CXCL14 directly into the TME could be an alternative approach to induce adaptive antitumor immune responses in combination with other immune-modulating therapies (e.g., a tumor vaccine). Previous studies have shown that delivering HPV epitope vaccines results in tumor suppression [103,104]. Thus, we hypothesize that combinations of CXCL14 with the HPV vaccines will further augment antitumor responses by enhancing T-cell infiltration and MHC-I antigen presentation, leading to robust tumor suppression [41,42]. The upregulation of MHC-I expression and the presentation of epitopes from the HPV vaccines could also enhance the efficacy of ICI therapy, establishing CXCL14 as an immunotherapeutic agent for effective strategies to treat HPV+ HNSCC. Figure 4 illustrates a hypothetical mechanism by which the combination of CXCL14 and HPV epitopes drives antitumor immunity in the context of HPV+ HNSCC.

## 9. Conclusions 

Our and others’ studies strongly suggest that CXCL14 plays an important role in activating antitumor immunity in multiple cancers, including HPV+ HNSCC. While CXCL14 could be used as a monotherapy, it is more likely to work in combination with other immunotherapy, such as ICIs. Nevertheless, using CXCL14 as an immunotherapeutic agent is still challenged by limitations in understanding its antitumor mechanisms, developing delivery methods, and its short half-life. Thus, further investigation and development of CXCL14 are necessary.

## Figures and Tables

**Figure 1 viruses-16-00302-f001:**
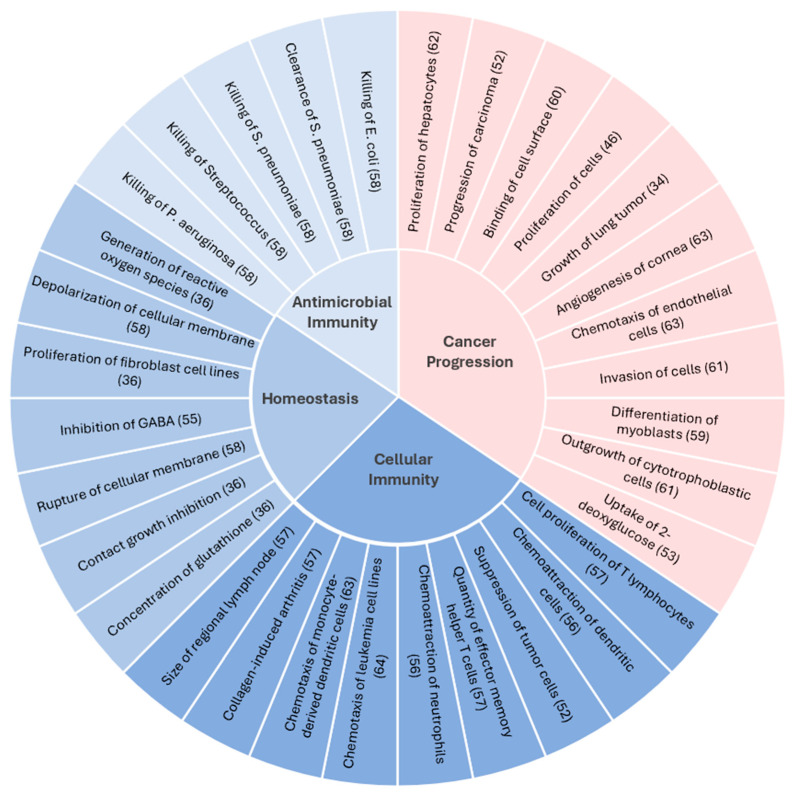
**Classification of CXCL14 gene-to-functions**. The functions and pathways associated with CXCL14 expression were analyzed using QIAGEN Ingenuity Pathway Analysis (IPA) (QIAGEN Inc. Redwood City, CA, USA, https://digitalinsights.qiagen.com/IPA (accessed on 21 May 2023)) and plotted in a colored circle. Blue and red segments correspond to increased and decreased functions and pathways relative to increased CXCL14 expression, respectively. The articles were collected from IPA analysis and then categorized into four broad groups (center pie chart) [34,36,46,52,53,55,56,57,58,59,60,61,62,63,64].

**Figure 2 viruses-16-00302-f002:**
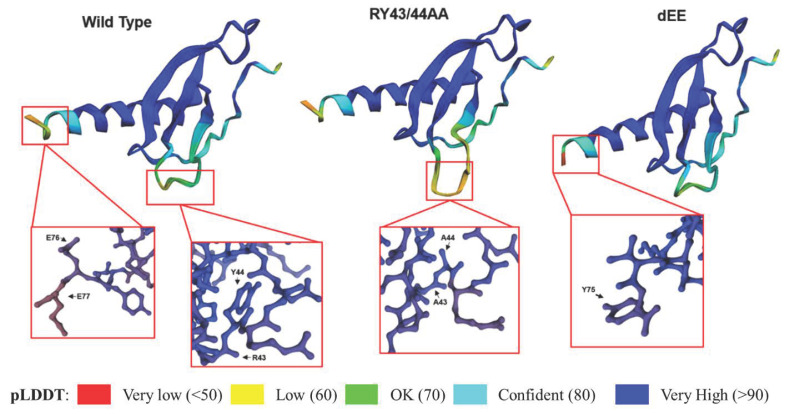
**In silico structural analysis of wildtype and mutant CXCL14 proteins.** Predicted folding representation of wildtype CXCL14, CXCL14-RY43/44AA, and CXCL14-dEE proteins were generated in ColabFold (v. 1.5.2), based on AlphaFold2 [91,92]. Full protein structures are shown, along with detailed representations of mutated regions (red boxes) generated in SWISS-MODEL Workspace [93,94,95,96,97]. AlphaFold-predicted local distance difference test (pLDDT) represents a per-residue confidence score for each amino acid in the full peptide corresponding to the colored key.

**Figure 3 viruses-16-00302-f003:**
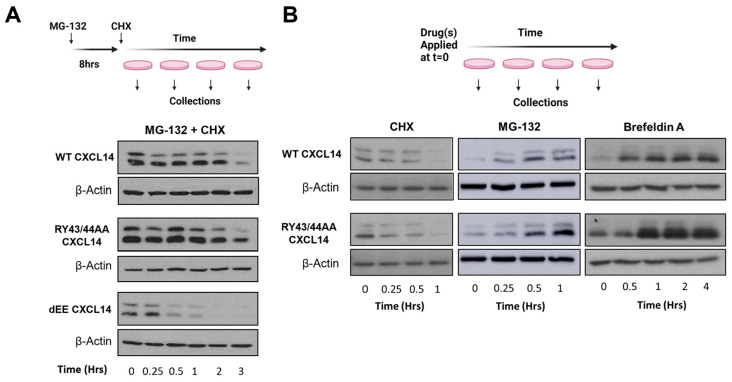
**Protein stability of wildtype and mutant CXCL14 proteins**. (**A**) Western blots with whole cell lysate prepared from 293T cells transiently transfected with wild type CXCL14 (CXCL14-WT), CXCL14 with RY43/44AA substitution (CXCL14-RY43/44AA), or CXCL14 with the deletion of two glutamates at the C-terminus (CXCL14-dEE). After 24 h, cells were treated with MG-132 (10 µM) for 8 h and CHX (50 µg/mL) as previously described [99]. (**B**) Western blots with whole cell extract prepared from 293T cells transiently transfected with CXCL14-WT or CXCL14-RY43/44AA. After 24 h, cells were treated with MG-132, CHX, or Brefeldin A (5 µg/mL) as previously described [99]. Experimental diagrams created with BioRender.com.

**Figure 4 viruses-16-00302-f004:**
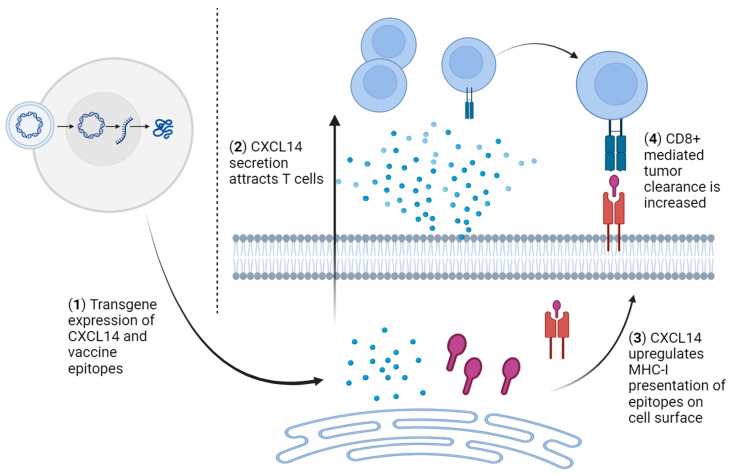
Schematic strategy for activating antitumor CD8+ T-cell responses using CXCL14 combined with therapeutic HPV vaccines. (1) CXCL14 and T-cell epitope peptides are expressed in tumor cells. (2) The secretion of CXCL14 attracts T cells into the TME. (3) CXCL14 upregulates the presentation of the epitope peptides by MHC-I. (4) Increased CD8+ T-cell infiltration and MHC-I upregulation synergistically enhance tumor cell clearance by cytotoxic CD8+ T cells. Created with BioRender.com.

## Data Availability

The primary uncropped images and gel blots can be found online at MDPI.

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
