# Peer review of "The Chemokine CXCL14 as a Potential Immunotherapeutic Agent for Cancer Therapy"

_viruses, 2024, doi:10.3390/v16020302_

Round 1
Reviewer 1 Report
Comments and Suggestions for Authors
Overall it is a very nicely written opinion. However, in some parts it is not clear waht the cited evidence for the relevance/function is. e.g. in the part "understanding the mechanism" murine and human in vitro as well as ex vivo and in vivo studies are mixed and sometimes the reader might imaging it as clinical trials, but after checking the references it becomes evident that only ex vivo oer even pure cell culture experiments have been performed.
Maybe it would also be good to write a paragraph to summarize shortly the hurdles to be tackled before applying CXCL14 as cancer therapy.
Finally, the conclusion is honest, but also weaker the the paper title suggests, so the paper title should be adapted.
Author Response
Overall it is a very nicely written opinion. However, in some parts it is not clear waht the cited evidence for the relevance/function is. e.g. in the part “understanding the mechanism” murine and human in vitro as well as ex vivo and in vivo studies are mixed and sometimes the reader might imaging it as clinical trials, but after checking the references it becomes evident that only ex vivo oer even pure cell culture experiments have been performed.
Response: We thank the reviewer for their comments and insights. We agree with the reviewer’s evaluation of the text regarding their first point. Our objective is to provide references to update readers on what is currently known regarding the mechanisms of CXCL14-mediated antitumor activity.
Maybe it would also be good to write a paragraph to summarize shortly the hurdles to be tackled before applying CXCL14 as cancer therapy.
Response: We appreciate the reviewer’s suggestion and view the hurdles to be overcome before applying CXCL14 as a cancer therapy as the crux of our opinion article. In the Abstract, we summarize that the current limitations for the potential therapeutic development of CXCL14 are the lack of understanding of what mechanism(s) govern CXCL14-mediated antitumor activity, how to deliver CXCL14, what dose to apply, and what combinations with existing therapy may boost antitumor immune responses in cancer patients. We also address these hurdles in their respective paragraphs. For example, on page 5 paragraph 2, the paragraph entitled “Understanding the mechanism of CXCL14-mediated tumor suppression” provides updates and references about what is known regarding CXCL14’s antitumor mechanism. Additionally, paragraph 2 on page 7, entitled: “CXCL14 therapeutic dose and combination therapy,” specifically addresses the hurdle of understanding what dose could be needed for a potential CXCL14 cancer therapy. Our description with the references cited in each section updates the current understanding of these limitations and provides insight into further investigations and developments in using CXCL14 as a cancer therapy.
Finally, the conclusion is honest, but also weaker the the paper title suggests, so the paper title should be adapted.
Response: We agree with the reviewer. As suggested, we have modified the title to “The chemokine CXCL14 as a potential immunotherapeutic agent for cancer therapy.”

Reviewer 2 Report
Comments and Suggestions for Authors
Since authors are working on CXCL14, Authors opinion do matter to the field. It is well written article but missing key component and experiments. Since author has worked to make more stable version of CXCL14, and suggests that more stable versions may have significant impact on tumor growth inhibition. It will be quite interesting to see the effect more stable version of CXCL14RY43/44AA has on tumor growth in vivo. Author has done these types of experiments in their previous publication “Suppression of Anti-tumor Immune Responses by Human Papillomavirus through Epigenetic Downregulation of CXCL14” and should not be difficult for the to do In vivo experiments to
Experiment for tumor growth monitoring with CXCL14 (CXCL14-WT), CXCL14 with RY43/44AA substitution (CXCL14-RY43/44AA), and (CXCL14-dEE), is suggested which will elevate the level of publication and give practical dimension to the opinion and will be a good contribution to the medical science.
Author Response
We thank the reviewer for their response and agree that tumor studies to increase CXCL14 protein stability are likely to be valuable in developing CXCL14 as a therapeutic agent. While we are planning to pursue the goal in the future, this Opinion article seeks to propose the possibility of altering CXCL14 protein stability as an important part of its therapeutic development.

Reviewer 3 Report
Comments and Suggestions for Authors
An interesting review on CXCL14, original, well written, well structured and illustrated.
Main recommendations:
-
Although the biology and research on CXCL14 is very well described here. In terms of discussion, the reasons why CXCL14 should be used rather than IL2 or other chemokines to enhance immunotherapeutics are not yet entirely clear.
It would be interesting to know or extend a bit more on other attempts to integrate this type of strategy into patient treatments and to highlight more why they think CXCL14 has the potential to do better than other chemokines CXCL11, IL2, IL7, IL15 etc… by exploiting their knowledge in terms of toxicity, intrinsic feature, efficacy, tumor type targeted.
Did the authors in their previous research make any comparisons in terms of functional readout ? Data to discuss on the disease of interest HPV?
2) Also, in the introductory section, it is felt that a little more on the potential toxicity of cytokine injections and the notion of dose would also be beneficial.
minor comments:
3) authors: “Thus, CXCL14 is an im-munotherapeutic agent for novel combination immunotherapy with an ICI, particularly for ICI nonresponders “ sounds too affirmative the way it is written. It's a hypothesis. wording to be adapted.
4) authors “Yet, given the global impact on DNA methylation by decitabine, treatment likely causes undesired ef-fects and/or drug-related toxicity in patients beyond restoring CXCL14 expression”
Authors should improve claity of the sentence by mentioning that many other genes are regulated by DNA methylation/global impact “on gene expression”…
5) missing literature
Ther Adv Med Oncol 2023 Dec 25:15:17588359231217966. doi: 10.1177/17588359231217966. eCollection 2023. CXCL14 as a potential marker for immunotherapy response prediction in renal cell carcinoma
Author Response
Main recommendations:
Although the biology and research on CXCL14 is very well described here. In terms of discussion, the reasons why CXCL14 should be used rather than IL2 or other chemokines to enhance immunotherapeutics are not yet entirely clear.
It would be interesting to know or extend a bit more on other attempts to integrate this type of strategy into patient treatments and to highlight more why they think CXCL14 has the potential to do better than other chemokines CXCL11, IL2, IL7, IL15 etc… by exploiting their knowledge in terms of toxicity, intrinsic feature, efficacy, tumor type targeted.
Response: We thank the reviewer for the thoughtful comments and insights. We agree and have updated the text to reflect the advantages of CXCL14 over other potential candidates (i.e., IL-2) in the paragraph “Developing CXCL14 as an immunotherapeutic agent” on page 4.
Did the authors in their previous research make any comparisons in terms of functional readout ? Data to discuss on the disease of interest HPV?
Response: Again, this is another good suggestion. We have not directly compared CXCL14 to other chemokines at this time. However, we will seek to directly compare the antitumor responses of other CXC chemokines with CXCL14.
2) Also, in the introductory section, it is felt that a little more on the potential toxicity of cytokine injections and the notion of dose would also be beneficial.
Response: We agree and have added “what dose to apply” in the Abstract. Additionally, we have also described what is known about the dose in the “CXCL14 therapeutic dose and combination therapy” section spanning pages 7 and 8. We believe that the references provide readers with information about what has been previously investigated regarding the dose and toxicity of cytokines.
minor comments:
3) authors: “Thus, CXCL14 is an immunotherapeutic agent for novel combination immunotherapy with an ICI, particularly for ICI nonresponders “sounds too affirmative the way it is written. It’s a hypothesis. wording to be adapted.
Response: We agree and have modified the text in the second to last sentence of the “Developing CXCL14 as an immunotherapeutic agent” paragraph which starts on page 4 to read: “Thus, CXCL14 has great potential as an immunotherapeutic agent for in novel combination immunotherapy with an ICI, particularly for ICI nonresponders.”
4) authors “Yet, given the global impact on DNA methylation by decitabine, treatment likely causes undesired effects and/or drug-related toxicity in patients beyond restoring CXCL14 expression”
Authors should improve claity of the sentence by mentioning that many other genes are regulated by DNA methylation/global impact “on gene expression”…
Response: We agree and have modified the text on page 5 under “Developing effective delivery tools for CXCL14” to read: “However, the expression of many genes is affected by DNA methylation.”
5) missing literature
Ther Adv Med Oncol 2023 Dec 25:15:17588359231217966. doi: 10.1177/17588359231217966. eCollection 2023. CXCL14 as a potential marker for immunotherapy response prediction in renal cell carcinoma
Response: We agree that this reference is relevant to our manuscript and have added to page 5 in the paragraph “Developing CXCL14 as an immunotherapeutic agent.”

Round 2
Reviewer 2 Report
Comments and Suggestions for Authors
Author has answered all relevant concerns, so paper can be accepted.